

# Biological control of important fungal diseases of potato and raspberry by two *Bacillus velezensis* strains

Anzhela Asaturova[1], Margarita Shternshis[1], Vera Tsvetkova[1,2], Tatyana Shpatova[1,2], Vladislava Maslennikova[2], Natalya Zhevnova[1] and Anna Homyak[1]

[1] Federal Research Center for Biological Plant Protection, Krasnodar, Krasnodar region, Russian Federation
[2] Novosibirsk State Agrarian University, Novosibirsk, Russian Federation

Corresponding author
Anzhela Asaturova,
asaturovaanzhela@yandex.ru

## ABSTRACT

Stem canker and black scurf caused by *Rhizoctonia solani* are the important diseases in potato, while spur blight caused by *Didymella applanata* is a major disease in red raspberry. In Western Siberia, both crops are grown predominantly in small-scale farming that requires maximal usage of biological products for plant protection instead of chemicals. We evaluated two promising *Bacillus velezensis* strains BZR 336 g and BZR 517 isolated in the south of Russia (45°1′N, 38°59′E) for their biological control potentials against the potato and red raspberry diseases under the more severe weather conditions of Western Siberia (55°1′N, 82°55′ E). We tested two techniques to apply biocontrol agents: (1) coating the seeds (potato tubers) and (2) spraying over the plants (raspberry canes). In each case, we estimated *B. velezensis* strains on two plant cultivars differed by the disease resistance. The degree of *B. velezensis* influence on disease incidence and severity depended on the bacterial strain, the protected plant, and its cultivar. We also demonstrated that two *B. velezensis* strains significantly stimulated plant growth of potato, which contributed to the plant productivity on both cultivars. The BZR 336 g strain affected the potato productivity more than the BZR 517 strain. Under the influence of both bacterial strains, raspberry yield was significantly higher compared to the control on the susceptible cultivar. These findings indicated that two southern *B. velezensis* strains had proved their efficacy as biological control agents in the control of the serious fungal infection of potato and raspberry plants under the more severe ecological conditions of Western Siberia. For the first time, we demonstrated *B. velezensis* strains potential for use as biological control agents against *R. solani* on potato, and against *D. applanata* on red raspberry.

## INTRODUCTION

Public concern over the harmful effects of chemical fungicides on the environment and human health has emphasized the need for alternative, ecologically-safe methods for controlling plant disease. Current trends in sustainable agriculture are concerned with

reducing the use of chemical pesticides that negatively affect the soil and water environment, the quality of agricultural products, and, as a consequence, animal and human health. An alternative to chemical products is the use of microbial agents isolated from natural habitats for plant disease control (*Lugtenberg, 2018*). This is especially important for regions with a short growing season, such as Western Siberia, where it is impossible to fully eliminate the negative effects of chemicals. This region has a severe continental climate characterized by sharp changes in temperature and humidity throughout the year. There are few studies on plant disease biocontrol in the region. Existing research is typically related to the biocontrol agents' efficacy on potato (*Solanum tuberosum* L.) and berry crops, particularly, red raspberry (*Rubus idaeus* L.) (*Shternshis et al., 2006*; *Bakhvalov et al., 2015*; *Tsvetkova et al., 2016*). We studied these crops because potato is one of the most common food crops in Western Siberia for its resistance to unpredictable changes of weather. The red raspberry is also sufficiently resistant to unfavorable abiotic environmental factors, especially low temperatures, making it a good candidate for study. This berry crop can withstand temperatures of −30 °C and lower (*Nestby &Takeda, 2015*; *Bogomolova & Ozherelieva, 2016*; *Graham & Brennan, 2018*) making its cultivation especially important in areas with a severe climate and sudden temperature changes. Both potato and raspberry are rich in vitamins, microelements, and other biologically active compounds that are beneficial to human health (*Rao & Snyder, 2010*; *Zhang & Nair, 2010*; *Furrer, Chegeni & Ferruzzi, 2018*). Potato and red raspberry plants are predominantly cultivated on small-scale farms in Western Siberia focused on organic agriculture. This farming method requires the maximal usage of biological products instead of chemicals.

Potato productivity is reduced in different regions of the world due to crop damage by harmful microorganisms. *Rhizoctonia solani* Kuhn is the most destructive potato disease, causing stem canker and black scurf (*Woodhall et al., 2007*; *Daami-Remadi, Zammouri & Mahjoub, 2008*; *Das et al., 2014*; *Yang et al., 2017*). In Western Siberia, the damage caused to potatoes by *Rhizoctonia* disease can reach 67–70.8% (*Tomilova et al., 2020*). One of the most common raspberry fungal diseases is spur blight caused by *Didymella applanata* (Niessl) Sacc. (*Williamson & Hargreaves, 1981*; *Stevic, Pavlovic & Tanovic, 2017*; *Vojinović et al., 2019*). In Western Siberia, spur blight is the most serious red raspberry disease.

Biological controls meet the goals of organic farming (*Lynch et al., 2012*). Bacterial action by species from the *Bacillus* genus, primarily *Bacillus subtilis*, have effectively controlled a common potato disease caused by *R. solani* in a number of countries for a decade (*Kumar et al., 2013*; *Ben Khedher et al., 2015*; Dinu, Boiu-Sicuia & Constantinescu, 2019*). In Western Siberia, early-maturing potato cultivars are in high demand due to its short vegetative period. However, these plants are more susceptible to yield losses caused by *R. solani* than others in other maturity groups (to an average of 35%). Therefore, it requires biological protection since potatoes are produced predominantly by small scale farmers in Western Siberia.

Biological control of red raspberry disease caused by *D. applanata* has not been well-studied (*Shternshis et al., 2006*; *Rekanovic et al., 2012*). The use of biocontrol agents

against plant pathogens has several advantages including increased antifungal activity, stimulating plant growth, and mitigating abiotic stress (*Yang, Kloepper & Ryu, 2009*; *Maksimov et al., 2015*; *Jochum et al., 2019*) leading to improved plant health (*Van Loon, 2007*; *Pérez-García, Romero & Vicente, 2011*).

Studying beneficial microorganisms to create new biological formulations for plant protection is typically conducted under the same ecological and geographical conditions in which the microorganisms were isolated. These circumstances explain why the application of biological formulations in other climatic regions is not always successful. Thus, it is important to determine biocontrol agents' efficacy in geographically distant regions with different climatic conditions when assessing new strains (*Elad & Pertot, 2014*). We studied two perspective *Bacillus velezensis* strains, BZR 336 g and BZR 517 (*Garkovenko et al., 2020*; *Radchenko et al., 2020*); both were previously referred to as *Bacillus subtilis* (*Asaturova et al., 2015*). These strains were isolated in southern Russia under different environmental conditions. The climate in southern Russia is temperate, suggesting that it has a dry subtropical climate. The lowest temperature reaches 0.8 °C in January and the highest temperature (35 °C) is reached in July. Snow cover on the soil surface lasts 10–14 days at a height of 10–20 cm. The soils are leached chernozem. The climate of Western Siberia is characterized by long, cold winters and short summers. The lowest temperature reaches −50 °C in January and the highest temperature (38 °C) is reached in July. The snow cover on the soil surface lasts 148–170 days at a height of more than 50 cm. The maximum soil freezing depth is 257 cm. The soils are gray forest.

*B. velezensis* strains are known as biological control agents that suppress certain dangerous plant diseases caused by *R. solanacearum* on tomato (*Chen et al., 2020*), *Fusarium graminearum* on corn (*Wang et al., 2020*), and *Botrytis cinerea* on strawberry (*Toral et al., 2020*). In addition to biocontrol, *B. velezensis* strains may promote plant growth (*Fan et al., 2018*; *Myo et al., 2019*; *Torres et al., 2020*). To date, there is no research on *B. velezensis*' biocontrol of *R. solani* on potato or *D. applanata* on raspberry diseases. In this study, we examined the effect of *B. velezensis* BZR 336 g and *B. velezensis* BZR 517 strains isolated in southern Russia on suppressing *Rhizoctonia* disease on potato and spur blight on red raspberry in Western Siberia, which is a geographically distant region with much more severe climate conditions.

## MATERIALS & METHODS

### *Bacillus* strains used for experiments

The *B. velezensis* BZR 336 g (Assembly: GCF_009683125.1, GenBank: NZ_WKKU00000000.1) and *B. velezensis* BZR 517 (Assembly: GCF_009683155.1, GenBank: NZ_WKKV00000000.1) strains were originally isolated from the winter wheat rhizosphere in the Krasnodar region (45°1′N, 38°59′E). The strains belonged to the Federal Research Center for Biological Plant Protection's (St. Petersburg–Pushkin, Russia) "State Collection of Entomoacariphages and Microorganisms" bioresource collection (No. 585858). The material and technical facilities of the unique scientific installation «New generation technological line for developing microbiological plant protection products» of the Federal Research Center for

Biological Plant Protection's (St. Petersburg–Pushkin, Russia) were used in the research (https://ckp-rf.ru/usu/671367/).

The strains were stored at 4 °C in test tubes with potato-glucose agar (50% potato broth, 2% glucose, 2% agar). For each experiment, fresh liquid cultures were prepared on the original optimized nutrient medium in New Brunswick Scientific Excella E25 (Framingham, MA, USA) incubator shakers (180 rpm). The optimized culture medium was created using Czapek's medium for bacteria. Molasses was used as a carbon source and corn extract was used as a nitrogen source (*Asaturova et al., 2015*). The *B. velezensis* BZR 336 g culture was cultivated at 25 °C for 48 h, and the *B. velezensis* BZR 517 culture was cultivated at 30 °C for 36 h. These parameters were obtained from a previous study (*Sidorova et al., 2020*).

### In vitro antifungal activity of *Bacillus* strains

We evaluated the antagonistic properties of *B. velezensis* strains against the phytopathogenic fungus *R. solani*, isolated from infected potato tubers, and against fungus *D. applanata*, isolated from infected raspberry canes in Western Siberia. *B. velezensis* BZR 336 g and *B. velezensis* BZR 517 strains were used at concentrations of $10^4$–$10^6$ CFU/mL. The experiment was conducted using the agar block techniques (*Shternshis, Shpatova & Belyaev, 2016*).

Strains' liquid cultures were introduced into potato glucose agar and Czapek medium for *R. solani* and *D. applanata*, respectively, at the indicated concentrations and poured into sterile petri dishes.

The solution was allowed to solidify before a 10 mm diameter agar block with a phytopathogenic fungus was placed in the center of the dish. The plates were incubated at 25 °C for 14 days in darkness. Petri dishes with a medium served as the control. Experiments were replicated five times and the entire experiment was conducted twice.

Antagonistic activity was evaluated by measuring the diameter of the fungal colonies after 3, 5, and 7 days. The fungal inhibition (I, %) was determined based on the data obtained:

$$I, \% = (Dc-Dt)/Dc \times 100$$

where Dc is the diameter of fungal colony in control and Dt is the diameter of fungal colony in treatment.

### Weather conditions in the summer months of 2015 and 2016

The average summer temperature in 2015 was 18.6 and 19.1 °C in 2016. The amount of precipitation for the same period in 2015 was 207 and 135 mm in 2016. The lowest precipitation occurred in June in 2015, and June and August in 2016. The wettest month was July in 2015. Higher temperatures were noted in July 2016 compared to July 2015.

### Field trials of *B. velezensis* strains on potato plants

Field trials were conducted on experimental plots at the Novosibirsk State Agrarian University (55°1′N, 82°55′E). Plant growth conditions included weeding, inter-row

cultivation, and hilling. We used Kemira potato complex fertilizer (Fertika, Russia) (50 g per m$^2$) consisting of nitrogen (11%), phosphorus (9%), potassium (17%), magnesium (2.7%), and sulfur (2.7%). We cultivated two potatoes from different maturity groups: early maturity cv. Yuna and mid-early cv. Svitanok kievskiy.

Each plot measured 30 m$^2$ and included 120 plants. Potato seed tubers were dipped for 30 min in a BZR 336 g or BZR 517 (10$^6$ CFU mL$^{-1}$) suspension prior to planting. The growing season was 3 months, during which the potato plant height, the number of above-ground stems, the biomass of daughter tubers, and potato yield were measured at 4, 6, and 10 weeks after planting. The experiment was arranged using a completely randomized block design. Four plot areas (10 m$^2$, 40 plants per plot) were subjected to each treatment. Tubers treated with water were used as control. Two independent experiments were performed. We did not observe any diseases except *R. solani*.

We assessed potato stems infected by *R. solani* and the plants' morphometric characteristics (height and number of stems per plant) 4, 6, and 10 weeks after planting. The disease incidence (DI%) was calculated by the formula: DI (%) = (T × 100)/N, where T is the number of infected plants and N is the total number of plants. Disease severity (DS, %) was estimated using the Townsend–Heuberger formula according to the following scale: 0 = no disease symptoms; 1 = stem canker up to 25 mm; 2 = stem canker up to 50 mm; 3 = stem canker more than 50 mm; 4 = canker surround the whole stem; five = stem completely nipped (*Frank, Leach & Webb, 1976*).

For daughter tubers the disease incidence (DI, %) was calculated using the formula:

$$DI(\%) = (T \times 100)/N$$

where T was the number of infected daughter tubers and N was the total number of daughter tubers.

We determined tubers' biomass distribution by their weight. The daughter tubers were categorized as: small (less than 35 g), medium (36–180 g), or large (more than 180 g). Yields were determined by weighing all daughter tubers from each plot.

## Field trials of *Bacillus* strains on raspberry plants

Raspberry plants were also represented by two cultivars: cv. Kirzhach is non-resistant to *Didymella applanata* and cv. Kolokolchik is resistant to this fungal pathogen.
The raspberry cultivar Kolokolchik was selected at the Siberian Research Institute of Horticulture (Barnaul, Russia), and the raspberry cultivar Kirzhach was selected from the All-Russian Institute of Horticulture and Nursery (Moscow, Russia). Raspberry plots were located at the Siberian Research Institute of Crop Production and Breeding's experimental station. Plant growth conditions included weeding and inter-row cultivation until the raspberry flowering stage. In these experiments, plots were not fertilized. During our study, we did not observe raspberry diseases other than *D. applanata*.

Plants were located in rows 40 cm wide; rows were spaced 2.5 m apart. The area of each plot was 10 m$^2$ and included 50 raspberry canes in four replicates. The size of the buffer zone between plots was 2 m. *B. velezensis* BZR 336 g and *B. velezensis* 517 strains were suspended at a concentration of 10$^6$ CFU mL$^{-1}$ and used to manage raspberry disease.

 

*B. velezensis* suspensions were applied at a volume application rate of 0.1 L/m$^2$ using a hand-held Orion-6 sprayer (Quazar Corp. Warsaw, Poland). The control plots were treated with water. Two independent experiments were performed. Bacterial strain treatment timing was determined at the appearance of the first disease symptoms. Observations were carried out throughout the growing season for 2 years.

Disease severity (DS, %) was estimated using the Townsend–Heuberger formula according to the following scale: 0 = healthy cane; 1 = patches with a smooth surface and a diameter less than 0.5 of the cane girth, appearance of splits was possible; 2 = patches with a smooth surface and the diameter more than 0.5 of the cane girth, appearance of splits was possible; 3 = patches with a deformed surface and the diameter more than 0.5 of the cane girth, splits or tissue deformation that come up to the vascular cylinder; 4 = patches with a deformed surface that completely cover the whole cane girth, splits or tissue deformation that come up to vascular cylinder (*Shternshis et al., 2002*). The disease incidence (DI, %) was calculated using the same formula used for potato plants.

A randomized complete block design was used to assign treatments to four replicates for both potato and raspberry cultures. We manually harvested the crops and the yields were determined by weighing all potato daughter tubers or all harvested raspberry fruits from each plot.

## Statistical analysis

We processed our data using Excel 2010 (Microsoft Corporation, Redmond, WA, USA). All results are expressed as the mean. We used STATISTICA 13.2 EN (trial version; Tibco, Palo Alto, CA, USA) to statistically analyze the data. Multiple comparisons of the means were performed using Duncan's tests with a significance level of $P = 0.05$.

# RESULTS

### Screening *B. velezensis* against *R. solani and D. applanata* in vitro

The studied strains inhibited the growth of *R. solani* and *D.applanata* under laboratory conditions (Fig. 1). The inhibition ratio depended on the concentration of bacteria in the nutrient medium (Tables 1 and 2). The progressive reduction of fungal growth was noted with increasing suspension concentration of *B. velezensis* strains. After 7 days, the diameter of the *R. solani* colonies under the action of two strains of *B. velezensis* significantly decreased at least five times ($P < 0.05$), and the percentage of the bacterial strains' inhibition activity at concentrations of 10$^5$ and 10$^6$ CFU/mL was rather high (Table 1). The same tendency was observed for *D. applanata* ($P > 0.05$) although the inhibitory effect was not as evident (Fig. 1; Table 2). In both cases, the inhibitory effect increased with time (Tables 1 and 2).

### Potato field trials

The disease incidence on potato stems in the control permanently increased during the growing seasons of in 2015 and 2016. The disease incidence in both years was no less than 80% on two potato cultivars in the control group 10 weeks after planting (Table 3). Using *B. velezensis* strains for 2 years reduced the disease incidence and severity on potato stems,

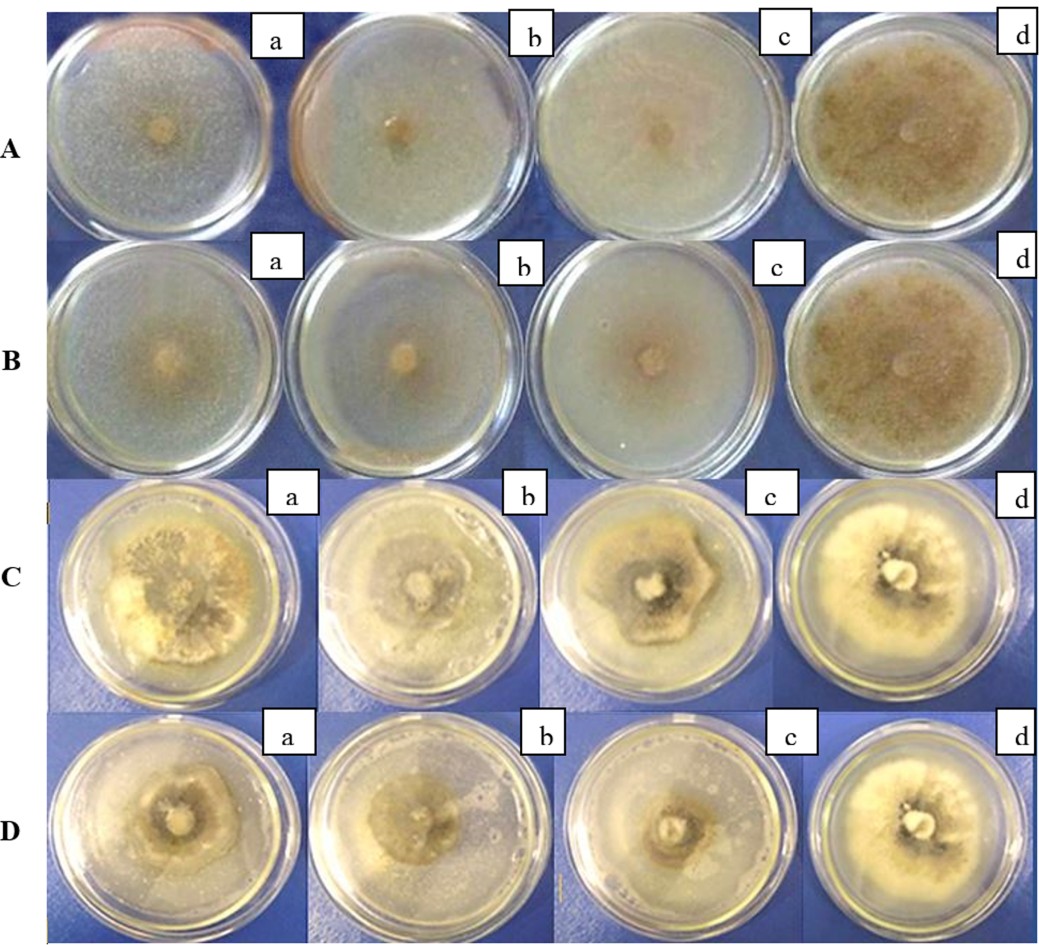

**Figure 1** (A, B) Antifungal activity of *B. velezensis* against R solani in vitro; (C, D) antifungal activity of *B. velezensis* against *D. applanata* in vitro. Concentration of bacterial suspension: a-104 CFU/mL, b-105 CFU/mL, c-106 CFU/mL, d–control.

**Table 1** The influence of *B. velezensis* strains on the growth of *R. solani* in vitro.

| *B. velezensis* strains | Colony diameter, cm, days | | | Inhibition of *R. solani*, %, days | | |
|---|---|---|---|---|---|---|
| | 3 | 5 | 7 | 3 | 5 | 7 |
| Control | 4.2c ± 0.43 | 7.9d ± 0.77 | 9.0d ± 0.00 | | | |
| BZR 336 g $10^4$ CFU/mL | 1.8ab ± 0.15 | 2.1ab ± 0.18 | 2.4b ± 0.31 | 57.1 | 73.4 | 73.3 |
| BZR 336 g $10^5$ CFU/mL | 1.6abc ± 0.11 | 1.7a ± 0.08 | 1.8a ± 0.08 | 61.9 | 78.5 | 80.0 |
| BZR 336 g $10^6$ CFU/mL | 1.4c ± 0.09 | 1.6a ± 0.18 | 1.7a ± 0.11 | 66.7 | 79.7 | 81.1 |
| BZR 517 $10^4$ CFU/mL | 1.9b ± 0.23 | 2.6c ± 0.58 | 2.7b ± 0.67 | 54.8 | 67.1 | 70.0 |
| BZR 517 $10^5$ CFU/mL | 1.5ac ± 0.11 | 1.7a ± 0.08 | 1.8a ± 0.04 | 64.3 | 78.5 | 80.0 |
| BZR 517 $10^6$ CFU/mL | 1.7ab ± 0.30 | 1.8ab ± 0.05 | 1.6a ± 0.09 | 59.5 | 77.2 | 82.2 |
| | $P = 0.06$ | $P = 0.01$ | $P = 0.00$ | | | |

**Note:**
Values followed by the same letter do not differ significantly at using Duncan's multiple-range test.

**Table 2 The influence of *B. velezensis* strains on the growth of *D. applanata* in vitro.**

| *Bacillus* strains | Colony diameter, cm, days | | | Inhibition of *D. applanata*, %, days | | |
|---|---|---|---|---|---|---|
| | 3 | 5 | 7 | 3 | 5 | 7 |
| Control | 3.2d ± 0.23 | 4.7e ± 0.19 | 6.4d ± 0.17 | | | |
| BZR 336 g $10^4$ CFU/mL | 2.4bc ± 0.13 | 4.1d ± 0.25 | 4.6c± 0.11 | 22.0 | 13.0 | 28.0 |
| BZR 336 g $10^5$ CFU/mL | 2.3ab ± 0.15 | 3.4a ± 0.15 | 3.9ab ± 0.28 | 28.1 | 27.6 | 39.0 |
| BZR 336 g $10^6$ CFU/mL | 2.5c ± 0.19 | 2.4c ± 0.12 | 3.9b ± 0.25 | 21.8 | 48.9 | 39.0 |
| BZR 517 $10^4$ CFU/mL | 2.3abc ± 0.19 | 3.7b ± 0.21 | 5.2c ± 0.28 | 28.1 | 21.3 | 18.7 |
| BZR 517 $10^5$ CFU/mL | 2.1a ± 0.11 | 3.5ab ± 0.29 | 4.6b ± 0.24 | 34.3 | 25.5 | 28.1 |
| BZR 517 $10^6$ CFU/mL | 2.2ab ± 0.09 | 3.6ab ± 0.16 | 4.6ab ± 0.31 | 31.2 | 23.4 | 28.1 |
| | $P = 0.36$ | $P = 0.35$ | $P = 0.11$ | | | |

**Note:**
Values followed by the same letter do not differ significantly at using Duncan's multiple-range test.

**Table 3 Effect of *B. velezensis* strains on disease incidence and severity caused by *R. solani* on two potato cultivars differed by disease resistance during 2015 and 2016 growing seasons in Western Siberia.**

| Cultivars | *Bacillus* strain | Year | Disease incidence, % weeks after planting | | | Disease severity, % weeks after planting | | | Disease incidence on the daughter tubers, % |
|---|---|---|---|---|---|---|---|---|---|
| | | | 4 | 6 | 10 | 4 | 6 | 10 | |
| Svitanok kievskiy | Control | 2015 | 21.5c ± 1.8 | 40.9c ± 2.37 | 94.4c ± 7.27 | 10.5c ± 0.48 | 17.3c ± 1.30 | 45.6b ± 5.26 | 50.6c |
| | BZR 336 g | | 0a ± 0 | 15.0a ± 1.01 | 11.1a ± 0.47 | 0a ± 0 | 3.5a ± 0.54 | 3.6a ± 0.39 | 14.4a |
| | BZR 517 | | 1.5b ± 0.05 | 33.3b ± 1.97 | 17.1b ± 0.55 | 0.8b ± 0.04 | 13.3b ± 1.08 | 4.6a ± 0.24 | 25.5b |
| | Control | 2016 | 11.2c ± 0.15 | 37.5c ± 2.19 | 83.3c ± 2.54 | 4.5c ± 0.57 | 10.0a ± 0.73 | 34.6c ± 2.16 | 86.7c |
| | BZR 336 g | | 3.9a ± 0.25 | 20.0a ± 1.54 | 26.7a ± 1.42 | 2.1a ± 1.64 | 5.7b ± 0.26 | 6.7a±0.67 | 5.6a |
| | BZR 517 | | 6.3b ± 0.25 | 23.3b ± 1.43 | 33.3b ± 2.64 | 3.0a ± 0.40 | 10.6b ± 0.89 | 9.3b ± 0.59 | 16.7b |
| Yuna | Control | 2015 | 30.1c ± 1.26 | 82.4c ± 3.48 | 83.3c ± 3.97 | 25.4c ± 1.27 | 28.2c ± 2.16 | 40.0c ± 2.33 | 40.4c |
| | BZR 336 g | | 14.1a ± 1.38 | 20.0a ± 1.34 | 26.1a ± 1.01 | 8.6b ± 0.63 | 8.6a ± 0.37 | 6.1a ± 0.42 | 16.1b |
| | BZR 517 | | 18.1b ± 0.80 | 29.4b ± 1.19 | 61.1b ± 2.40 | 7.4a ± 0.53 | 14.1b ± 0.91 | 23.3b ± 1.88 | 4.9a |
| | Control | 2016 | 8.3c ± 0.71 | 65.9c ± 4.81 | 80.0c ± 3.16 | 20.0c ± 0.78 | 23.2c ± 0.87 | 31.2c ± 2.68 | 76.7c |
| | BZR 336 g | | 4.7a ± 0.37 | 22.5a ± 1.82 | 30.0a ± 1.95 | 10.3a ± 0.94 | 6.0a ± 0.43 | 7.3a ± 0.32 | 23.3a |
| | BZR 517 | | 7.7b ± 0.53 | 34.0b ± 2.03 | 56.0b ± 4.39 | 14.9b ± 0.57 | 11.6c ± 1.10 | 18.4b ± 1.08 | 27.8b |

**Note:**
Means followed by the same letters in the same column are not significantly different ($P \leq 0.05$) according to Duncan's multiple-range test.

the degree of which depended on the strain, potato cultivar, and weather conditions that year. In 2015, a statistically significant decrease of the disease incidence ($P < 0.05$) was observed in both potato cultivars in all cases 6 weeks after planting (Table 3). The disease severity on the stems decreased by 79.8% under BZR 336 g ($P < 0.05$), and by 23.1% under BZR 517 ($P > 0.05$) on cv. Svitanok kievskiy, and cv. Yuna by 69.5% and 50%, respectively ($P < 0.05$) (Table 3). During the same period in 2016, a statistically significant decrease in disease severity of 74.1% was observed on cv. Yuna, under the action of the BZR 336 g strain and of 50% by BZR 517 ($P < 0.05$) (Table 3).

The greatest decrease in the disease' incidence and severity on the stems after application was achieved 10 weeks after treatment by bacilli in both 2015 and 2016. In 2015, 10 weeks after treating Svitanok kievskiy tubers with the *B. velezensis* BZR 336 g

**Table 4** Effect of *B. velezensis* strains on morphometric characteristics of two potato cultivars differed by disease resistance to *R. solani* during 2015 and 2016 growing seasons in Western Siberia.

| Cultivars | *Bacillus* strain | Year | Plant height, cm, weeks after planting | | | The number of stems per plant, weeks after planting | | |
|---|---|---|---|---|---|---|---|---|
| | | | 4 | 6 | 10 | 4 | 6 | 10 |
| Svitanok kievskiy | Control | 2015 | 11.6a ± 0.7 | 22.4a ± 1.4 | 31.1b ± 1.8 | 2.1a ± 0.2 | 4.4a ± 0.3 | 4.6b ± 0.4 |
| | BZR 336 g | | 11.8a ± 0.7 | 26.7b ± 1.7 | 40.1a ± 1.7 | 5.1c ± 0.5 | 7.0c ± 0.4 | 8.0a ± 0.4 |
| | BZR 517 | | 12.6a ± 0.7 | 24.7c ± 1.4 | 38.5a ± 2.0 | 3.7b ± 0.4 | 6.0b ± 0.4 | 7.1a ± 0.5 |
| | Control | 2016 | 24.0b ± 1.9 | 31.2a ± 2.1 | 38.2a ± 1.8 | 3.6a ± 0.4 | 4.4a ± 0.2 | 5.0a ± 0.6 |
| | BZR 336 g | | 30.0a ± 2.1 | 37.4b ± 2.1 | 41.2b ± 2.7 | 7.0c ± 0.4 | 7.0 c ± 0.6 | 8.0c ± 0.5 |
| | BZR 517 | | 28.0a ± 1.3 | 34.6ab ± 1.9 | 39.0ab ± 1.9 | 5.0b ± 0.4 | 6.0 b ± 0.5 | 6.0b ± 0.4 |
| Yuna | Control | 2015 | 11.6a ± 1.0 | 31.0a ± 1.5 | 31.2a ± 1.4 | 3.0a ± 0.2 | 3.4b ± 0.4 | 3.4b ± 0.2 |
| | BZR 336 g | | 15.9b ± 0.8 | 33.1b ± 1.4 | 43.2b ± 1.4 | 4.0c ± 0.2 | 4.6a ± 0.3 | 6.0a ± 0.6 |
| | BZR 517 | | 12.6a ± 0.9 | 30.2a ± 2.3 | 33.9a ± 1.5 | 3.5b ± 0.3 | 4.4a ± 0.3 | 5.4a ± 0.5 |
| | Control | 2016 | 19.0a ± 1.2 | 27.2b ± 1.6 | 31.0a ± 1.8 | 2.4a ± 0.2 | 3.4a ± 0.4 | 4.0a ± 0.3 |
| | BZR 336 g | | 28.0c ± 1.8 | 34.0a ± 1.7 | 41.4b ± 2.4 | 4.6c ± 0.3 | 5.0c ± 0.4 | 6.0b ± 0.4 |
| | BZR 517 | | 22.0b ± 1.7 | 32.2a ± 1.3 | 34.8a ± 1.7 | 3.8b ± 0.4 | 4.0b ± 0.3 | 4.4a ± 0.3 |

Notes:
Means followed by the same letters in the same column are not significantly different ($P \leq 0.05$) according to Duncan's multiple-range test.
Effect of *B. velezensis* strains on morphometric characteristics of two potato cultivars differed by disease resistance to *R. solani* during 2015 and 2016 growing seasons in Western Siberia.

strain, the disease incidence decreased by 71.5%, and on cv. Yuna by 61.1% ($P < 0.05$). In the same year, the strains' influenced on disease severity was even stronger ($P < 0.05$) (Table 3), and a smaller effect was observed on cv. Yuna. In 2016, with less air humidity, bacilli's effect on the disease incidence and severity was statistically smaller ($P < 0.05$) than in the previous year (Table 3).

The stem damage during the growing season in the control group resulted in a large number of daughter tubers infected by *R. solani* sclerotia. Damage was especially prominent in 2016, when the disease incidence in the control daughter tubers was 86.7% (Table 3). However, treating tubers with *B. velezensis* strains significantly reduced the disease incidence in daughter tubers. This value decreased by 71.5% in 2015 and by 93.5% in 2016 in cv. Svitanok kievskiy when using the BZR 336 g strain (Table 3). The BZR 517 strain decreased the disease incidence by 49.6% and 89.6%, respectively ($P < 0.05$). In cv. Yuna daughter tubers, the data for both strains differed from the control significantly in both years. In 2015, the strain BZR 517 was 3.3 times more effective compared to BZR 336 g. In 2016, BZR 517 was significantly less effective than BZR 336 g ($P < 0.05$) (Table 3).

The bacterial strains also contributed to the potato plant growth stimulation during the growing seasons. In 2015, the seed treatment led to significant increase in plant height ($P < 0.05$) in the most cases (Table 4). Under the influence of both strains, in cv. Svitanok kievskiy the number of stems per plant significantly increased in 4, 6, and 10 weeks after planting. A similar result was observed in cv. Yuna ($P < 0.05$) (Table 4). The effect of these strains on plant height in 2016 was less for cv. Svitanok kievskiy (Table 4). Ten weeks

**Table 5 Effect of *B. velezensis* strains on the biomass distribution of daughter tubers of two potato cultivars differed by disease resistance to *R. solani* and total yield during 2015 and 2016 growing seasons in Western Siberia.**

| Cultivars | *Bacillus* strain | Biomass distribution by fractions, % | | | | | | Yield, t ha$^{-1}$ | |
| --- | --- | --- | --- | --- | --- | --- | --- | --- | --- |
| | | 2015 | | | 2016 | | | 2015 | 2016 |
| | | Small | Medium | Large | Small | Medium | Large | | |
| Svitanok kievskiy | Control | 16.6c ± 0.7 | 64.1b ± 2.5 | 19.3b ± 0.6 | 16.9c ± 0.7 | 58.3b ± 1.9 | 24.8b ± 1.6 | 20.3a ± 1.2 | 11.6c ± 0.9 |
| | BZR 336 g | 8.4b ± 0.4 | 59.1a ± 1.3 | 32.5c ± 1.7 | 4.6a ± 0.2 | 52.8a ± 2.5 | 40.0c ± 1.8 | 28.7d ± 0.3 | 18.9a ± 1.1 |
| | BZR 517 | 11.8a ± 0.8 | 75.3a ± 2.4 | 12.9a ± 0.7 | 7.2b ± 0.4 | 71.7c ± 2.5 | 21.1a ± 1.3 | 22.0b ± 0.6 | 16.5b ± 0.7 |
| Yuna | Control | 13.1c ± 0.9 | 63.8a ± 4.3 | 23.1a ± 1.5 | 11.4c ± 1.0 | 67.8a ± 5.4 | 20.8b ± 1.3 | 21.0ab ± 0.7 | 15.5b ± 0.3 |
| | BZR 336 g | 3.8a ± 0.2 | 54.2b ± 2.3 | 42.0c ± 2.3 | 3.0a ± 0.4 | 66.7a ± 3.1 | 30.3a ± 1.8 | 25.7c ± 1.6 | 19.7a ± 0.7 |
| | BZR 517 | 6.0b ± 0.4 | 65.9a ± 4.5 | 28.1b ± 1.2 | 6.3b ± 0.6 | 64.8a ± 3.6 | 28.9a ± 1.4 | 24.6c ± 1.3 | 19.6a ± 1.1 |

**Note:**
Means followed by the same letters in the same column are not significantly different ($P ≤ 0.05$) according to Duncan's multiple-range test.

after planting, treating the two potato cultivars with both strains over 2 years led to a significant increase in the morphometric characteristics of the plants (Table 4).

The pre-planting treatment of potato tubers with the bacterial strains had a positive effect on the quality of the daughter tubers. Thus, the effect of the strain BZR 336 g appeared to increase with the increasing biomass of the tubers. In 2015, the effect was increased by 37.2% and by 45.0% in cv. Svitanok kievskiy and cv. Yuna, respectively (Table 5). In 2016, the biomass was increased by 31.3–38.0% in both cultivars (Table 5). BZR 517 caused an increase in the medium fraction of tubers (Table 5). As a result, the use of the BZR 336 g strain provided a significant increase of 41.4% in daughter tubers' biomass in the mid-early cv. Svitanok kievskiy in 2015. In 2016, biomass increased by 62.9%. The biomass was increased in early cv. Yuna by 22.4% and by 27.4% ($P < 0.05$), respectively (Table 5). Tubers pre-treated with the BZR 517 strain contributed less to the yield formation. In 2015, the daughter tubers's biomass increased by 10% (cv. Svitanok kievskiy) and by 17.1% (cv. Yuna), respectively (Table 5). In 2016, the tuber biomass increased by 42.2% (cv. Svitanok kievskiy) and by 26.5% (cv. Yuna) compared with the control ($P < 0.05$) (Table 5). The total potato yield was significantly increased in both cultivars compared to the control ($P < 0.05$) (Table 5).

## Red raspberry field trials

Raspberry cane disease incidence increased in the control differently during the 2 year study period. This increase was more serious during the 2015 season than in 2016 (Table 6) due to increased precipitation in 2015 (207 mm) compared to 2016 (135 mm).

*B. velezensis* strains were able to reduce the disease incidence. In 2015, we observed a 69.3% ($P < 0.05$) reduction in disease incidence 4 weeks after BZR 517 treatment on cv. Kirzhach; in cv. Kolokolchik, the reduction was 50.0% ($P > 0.05$). In 12 weeks, *B. velezensis* significantly ($P < 0.05$) reduced this value by 46.4% in cv. Kirzhach, and by 59.7% in cv. Kolokolchik (Table 6). In 2016, 4 and 12 weeks after the treatment both strains significantly reduced disease incidence ($P < 0.05$) to the same degree in cv. Kirzhach (Table 6). The disease incidence was reduced by more than 50% at 4 and 12 weeks in

**Table 6 Effect of *B. velezensis* strains on disease incidence and severity caused by *D. applanata* on two raspberry cultivars differed by disease resistance and total yield during 2015 and 2016 growing seasons in Western Siberia.**

| Cultivar | *Bacillus* strain | Year | Disease incidence, % weeks | | Disease severity, % weeks | | Yield, t ha$^{-1}$ |
|---|---|---|---|---|---|---|---|
| | | | 4 | 12 | 4 | 12 | |
| Kirzhach | Control | 2015 | 32.5c ± 5.0 | 70.0d ± 8.2 | 8.1b ± 1.3 | 20.5b ± 3.2 | 3.30b ± 0.09 |
| | BZR 336 g | | 15.0b ± 5.8 | 40.0c ± 8.2 | 3.8a ± 1.4 | 10.0a ± 2.0 | 3.50a ± 0.02 |
| | BZR 517 | | 10.0ab ± 0 | 37.5bc ± 5.0 | 2.5a ± 0.0 | 9.4a ± 1.3 | 3.52a ± 0.01 |
| | Control | 2016 | 42.5b ± 12.6 | 52.5c ± 5.0 | 10.6b ± 0.6 | 19.4b ± 3.8 | 1.32c ± 0.03 |
| | BZR 336 g | | 12.5a ± 5.0 | 42.5ac ± 9.6 | 3.1a ± 0.7 | 11.3a ± 1.2 | 1.37a ± 0.02 |
| | BZR 517 | | 12.5a ± 5.0 | 37.1a ± 8.8 | 3.1a ± 0.7 | 9.3a ± 1.6 | 1.41b ± 0.01 |
| Kolokolchik | Control | 2015 | 9.2ab ± 0.7 | 30.0b ± 0.0 | 2.3a ± 0.5 | 7.5b ± 2.2 | 3.61a ± 0.01 |
| | BZR 336 g | | 3.8a ± 4.4 | 15.9a ± 4.1 | 1.0a ± 1.2 | 4.0a ± 0.5 | 3.62a ± 0.00 |
| | BZR 517 | | 4.6a ± 3.5 | 12.1a ± 4.3 | 1.2a ± 1.4 | 3.0a ± 0.2 | 3.61a ± 0.00 |
| | Control | 2016 | 15.0a ± 12.9 | 32.5a ± 9.6 | 3.8a ± 3.2 | 8.1b ± 2.4 | 2.59a ± 0.02 |
| | BZR 336 g | | 7.5a ± 5.0 | 16.3b ± 4.2 | 1.9a ± 2.4 | 4.1a ± 1.1 | 2.60ab ± 0.01 |
| | BZR 517 | | 6.3a ± 4.4 | 15.0b ± 5.8 | 1.6a ± 1.2 | 3.8a ± 1.4 | 2.60ab ± 0.01 |

Note:
Means followed by the same letters in the same column are not significantly different ($P \leq 0.05$) according to Duncan's multiple-range test.

cv. Kolokolchik under the two treatment strains. This value was only significant at 12 weeks ($P < 0.05$).

In 2015, 4 weeks after treating the raspberry plants, we did not find a statistically significant difference ($P > 0.05$) in disease severity after both treatments on cv. Kirzhach (Table 6). The tendency in this reduction was observed only in the resistant cv. Kolokolchik ($P > 0.05$). A significant reduction in disease severity was observed using the *B. velezensis* 517 strain (69.1% on cv. Kirzhach and 47.8% on the resistant cv. Kolokolchik ($P < 0.05$)) (Table 6). Spur blight severity decreased by 54.1% in cv. Kirzhach, and 60% in cv. Kolokolchik ($P < 0.05$) compared to the control.

In 2016, the disease severity on the raspberry plant cv. Kirzhach significantly decreased by 70.8% due to both strains 4 weeks after treatment and by 52.1% 12 weeks after treatment. The disease severity significantly decreased by almost 50% in 12 weeks using the more resistant cv. Kolokolchik ($P < 0.05$) (Table 6). Fruit yield increased significantly in the cv. Kirzhach under the influence of both strains in 2015 and 2016 ($P < 0.05$) (Table 6), but there was no increase in this value in the cv. Kolokolchik in these years. This variation could be due to factors such as cvs. Kirzhach and Kolokolchik's susceptibility to the pathogen or to different weather conditions in 2015 and 2016.

## DISCUSSION

We evaluated using two *B. velezensis* strains as biological control agents to address the most common fungal diseases affecting potato and red raspberry in Western Siberia. We assessed the antagonistic activity of these strains, which were isolated from soil in southern part of Russia (Krasnodar), against *R. solani* on potato and against *D. applanata* on raspberry in Western Siberia. It should be noted that Krasnodar is an important

agricultural region with a growing period of 165 to 200 days. In Western Siberia, the growing season is approximately 115–130 days.

Certain *Bacillus* strains have been shown to effectively control *R. solani* disease on potato in different geographic regions around the world (*Saber et al., 2015*; *Dinu, Boiu-Sicuia & Constantinescu, 2019*; *Larkin, 2020*). However, no studies have been conducted using *B. velezensis* for this purpose.

We revealed the ability of the studied strains to suppress *Rhizoctonia* disease for two potato cultivars (early maturity cv. Yuna and mid-early maturity cv. Svitanok kievskiy) in Western Siberia using field trials for 2 years. The conditions differed in terms of temperature and environmental humidity. We showed the ability of *B. velezensis* strains isolated in the warm region south of Russia to remain active against *R. solani* in the more severe climactic conditions of Western Siberia. The *Bacillus* species are known to inhibit the growth of *R. solani* by producing lipopeptides, chitinases, surfactins and other biologically active metabolites (*Saber et al., 2015*; *Shafi, Tian & Ji, 2017*). In addition, some authors have demonstrated the ability of *B. velezensis* strains to stimulate plant growth (*Ye et al., 2018*; *Dinu, Boiu-Sicuia & Constantinescu, 2019*; *Myo et al., 2019*). The results of our 2-year research showed that, along with the antifungal effect, the bacilli stimulated plant growth by increasing parameters such as plant height and number of stems per plant. In addition, data presented in our study are in accordance with other research indicating that some *Bacillus* strains offer additional benefit to the plants, such as increased tolerance to abiotic stress and growth promotion (*Enebe & Babalola, 2018*; *Shameer & Prasad, 2018*; *Jochum et al., 2019*). Our results also correspond to other research on two *Bacillus* strains that protected potato plants against abiotic stress (*Vurukonda et al., 2016*).

We indicated that the two *B. velezensis* strains' efficacy against *R. solani* was dependent on the potato cultivar. According to the several parameters, such as disease incidence and severity on the potato stems and daughter tubers, plant growth stimulation, and daughter tubers' biomass, the response of potato plants of mid-early maturity cv. Svitanok kievskiy to the bacilli treatment was more pronounced compared with the early maturity cv. Yuna. It should be noted that we used two *B. velezensis* strains at a concentration of $10^6$ CFU mL$^{-1}$ to control *R. solani* which were less than typically used in research.

A positive effect of the studied *B. velezensis* strains resulted in increased potato crop productivity. Decreased potato yield in 2016 compared to 2015 should be explained by the drought during the period of active tuberization. June and early July 2016 were dry, and only 165.7 mm of rainfall was observed during the vegetation period, which is 1.7 times less than long-term annual data. However, it was excessively wet in July 2015. In general, the strain BZR 336 g contributed more significantly to the tuber yield.

We also compared the effectiveness of these two *B. velezensis* strains on red raspberry plants that differed by plant biology and other biological features. The results of field testing two bacterial strains on red raspberry plants revealed their effectiveness against spur blight depended both on the properties of the bacterial strain and on the raspberry cultivar. In 2015 and 2016, the increase in disease incidence and severity on raspberry canes was also demonstrated during the growing season in control. We observed some

distinction between the two cultures in addition to the similar tendency in the influence of bacterial strains on two different plants. The disease suppression was significantly more pronounced under *B. velezensis* BZR 336 g in potatoes. However, *B. velezensis* BZR 517 and *B. velezensis* BZR 336 g performed almost similarly against raspberry spur blight. The raspberry yield slightly increased after bacilli treatment on cv. Kirzhach but this was not reflected in the resistant cv. Kolokolchic.

It is also well known that modes of action of biocontrol agents include induced systemic resistance (ISR). Direct antifungal effect, plant growth promotion, and enhanced tolerance to abiotic stress may be due to the mechanism of ISR (*Kloepper, Ryu & Zhang, 2004*; *Van Loon, 2007*; *Pieterse et al., 2014*; *Kohl, Kolnaar & Ravensberg, 2019*). ISR often aids *B. velezensis* in protecting against phytopathogens (*Chen et al., 2018*; *Grady et al., 2019*; *Rabbee et al., 2019*; *Xie et al., 2019*). ISR initiation requires beneficial microbes to efficiently colonize the root systems of host plants. This was confirmed by applying *Bacillus* strains to various plants (*Pieterse et al., 2014*). However, there is less research showing ISR's possible contributions to the treatment of plants' above grounds parts. Thus, we should assume that *B. velezensis*'s contributions to the ISR mechanism in potato may be greater than in raspberry, which was confirmed by their influence on crop yields.

According to our data, the *B. velezensis* strains isolated in a favorable agroecological environment demonstrated that they are effective under more severe ecological and geographical conditions and can managing potato and raspberry plant health in Western Siberia. This was dependent on the bacterial strain, the protected plant, the plant cultivar, and on the pathogen causing the plant disease.

We also demonstrated that two *B. velezensis* strains stimulated the growth of the potato plant, which contributed to the plant productivity. It should be noted that biocontrol methods are very promising sources of plant disease control in organic farming (*Lindsey, Murugau & Rentita, 2020*). Some *B. velezensis* strains are used in organic agriculture as an alternative to chemical pesticides (*Charon-Lamoureux et al., 2020*). Organic farming systems are more profitable and environmentally friendly; they deliver foods that contain less (or no) pesticide residues when compared with conventional farming (*Reganold & Wachter, 2016*). This is important for potato and raspberry planted in small farms. According to *Jouzi et al. (2017)*, organic farming may present some significant challenges to small-scale farmers.

## CONCLUSIONS

We showed that two southern *B. velezensis* strains were effective as biological control agents in the control of the serious fungal infection of potato and raspberry plants in the field under the more severe ecological conditions than those from which they were isolated. Overall, the use of two *B. velezensis* strains on potato and red raspberry cultivars under conditions of Western Siberia improved plant health and may protect the environment. The strains studied are the basis for developing new biocontrol formulations suitable for applications in geographic locations with highly different environmental conditions. Their use corresponds with the aims of sustainable agriculture. The use of biocontrol agents as an alternative to chemical fungicides is in line with the goals of

sustainable horticulture. We demonstrated the novel potential use of *B. velezensis* strains as biological control agents against *R. solani* on potato plants and against *D. applanata* on red raspberry plants.

### Funding

The studies were carried out in accordance with State Assignment No. 075-00376-19-00 of the Ministry of Science and Higher Education of the Russian Federation as part of research on the topic No. 0686-2019-0013 and was supported by the Russian Foundation for Basic Research (Project No. 13-08-96533). There was no additional external funding received for this study. The funders had no role in study design, data collection and analysis, decision to publish, or preparation of the manuscript.

### Grant Disclosures

The following grant information was disclosed by the authors:
Ministry of Science and Higher Education of the Russian Federation: 075-00376-19-00 and 0686-2019-0013.
Russian Foundation for Basic Research: 13-08-96533.

### Competing Interests

The authors declare that they have no competing interests.

### Author Contributions

- Anzhela Asaturova conceived and designed the experiments, analyzed the data, authored or reviewed drafts of the paper, and approved the final draft.
- Margarita Shternshis conceived and designed the experiments, analyzed the data, prepared figures and/or tables, authored or reviewed drafts of the paper, and approved the final draft.
- Vera Tsvetkova performed the experiments, prepared figures and/or tables, and approved the final draft.
- Tatyana Shpatova performed the experiments, prepared figures and/or tables, authored or reviewed drafts of the paper, and approved the final draft.
- Vladislava Maslennikova performed the experiments, prepared figures and/or tables, and approved the final draft.
- Natalya Zhevnova performed the experiments, analyzed the data, authored or reviewed drafts of the paper, and approved the final draft.
- Anna Homyak performed the experiments, authored or reviewed drafts of the paper, and approved the final draft.

### DNA Deposition

The following information was supplied regarding the deposition of DNA sequences:
Assembly: GCF_009683125.1 is available in GenBank: NZ_WKKU00000000.1
Assembly: GCF_009683155.1 is available in GenBank: NZ_WKKV00000000.1

## Data Availability

The raw measurements are available in the Supplemental File.

## Supplemental Information

Supplemental information for this article can be found online at http://dx.doi.org/10.7717/peerj.11578#supplemental-information.

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
