# Peer review of "Biological control of important fungal diseases of potato and raspberry by two Bacillus velezensis strains"

_PeerJ, doi:10.7717/peerj.11578_

## Round 0.1 · original submission · Major Revisions

Dear authors, I would like to see you read and carefully incorporate the changes suggested by the reviewers, in particular Reviewer 2. I think that these changes will improve your manuscript considerably. Also, reviewers are concerned about the organizational logic of the different ideas presented and so a serious re-organization of the writing is also imperative. I will be glad to see your revised manuscript in the near future.

Reviewer 1 ·

Basic reporting

English language used in the manuscript seems to be clear, professional and understandable.
Literature is well referenced and relevant.
Structure conforms to PeerJ standards.
Figures are relevant, well labelled and described. Figure S2 is high quality. Figure S1 seems to have too low resolution.
Custom check. I have successfully downloaded all the files mentioned in the article. I also have successfully performed DNA data checks.
Can I access the deposited data? – Yes.
Has the data been deposited correctly? – Yes.
Is the deposition information noted in the manuscript? – Yes.

Experimental design

The results presented in the manuscript are original primary research within the scope of the journal.
Research questions are well defined, relevant and meaningful from the point of biological control of phytopathogenic fungi Rhizoctonia solani and Didymella applanata. The research fills knowledge gaps regarding the use of Bacillus for biological control of phytopathogenic fungi in various environmental conditions. The authors have demonstrated the effectiveness of southern bacterial strains as biological control agents for fungal infections of potatoes and raspberries in more severe environmental conditions. The results are valuable both for agricultural biotechnology and for microbial ecology.
The study adheres to technical and ethical standards.
Methods are described with sufficient details and information to replicate, with some exceptions (see comments below).

Validity of the findings

All underlying data have been provided; they are robust, statistically sound and controlled.
Conclusions are well stated, linked to original research question and limited to supporting results.

Additional comments

Line 129. Error in the text. 104-106 CFU/mL instead of exponent.
Lines 153-154. Insufficient information about the fertilizer (how much nitrogen, phosphorus, potassium, magnesium, and sulfur is in its composition)?
Lines 236-241, 245, 246. Do the authors mean percentages or percentage points? I would recommend use both percentages and percentage points.
Statistical approach. In the future, I would recommend that authors pay special attention to multivariate statistics (Principal components analysis, Discriminant analysis and other very useful methods for analysis multivariate data).

Reviewer 2 ·

Basic reporting

- There are some problems in the English language throughout the manuscript.
- The study presents interesting data and has the potential for being an important contribution regarding biological control after being revised.
- Some references in the text are missing and they were pointed out in my revision.
- Too many tables (9) are presented.

Experimental design

Please check the general comments for the author

Validity of the findings

.

Additional comments

The manuscript “Biological control of important fungal diseases of potato and raspberry by two Bacillus velezensis strains” investigated the biological control potential of these B. velezensis strains against fungal diseases in potato and red raspberry cultivated in Russia. The study presents interesting data and has the potential for being an important contribution regarding biological control. However, the manuscript provides an insufficient description of the used methods and the data analysis presented in the manuscript should be improved in order for the manuscript to reach its full potential. The English language should be corrected in different parts of the manuscript. My detailed comments regarding my assessment of the presented manuscript can be found below. I hope that my comments will be constructive and that they will help to improve the manuscript.
Introduction
Line 53 – Explain (or provide a reference) why the full restoration of nature from the negative effects of chemicals is impossible in Western Siberia. There are other places worldwide that short growing seasons occur. Is this true or not?
Line 54 – Can you give an estimative of the sharp changes in temperature and humidity during a year?
Line 79 - by using biological control that corresponding conception of organic agriculture – do you mean “that corresponds…”? Biological control fits into the concept of organic farming – is that you mean?
Line 81 - Biological control of a common potato disease caused by R. solani, have been used by bacteria of the genus Bacillus… Biological control has been used? Other meaning? Please rewrite the sentence.
Line 82 - Biological control of red raspberry disease caused by D. applanata, was much less known? It was less known but now it is not anymore? Please rewrite the sentence.
Line 98 - in another locality with different environmental conditions. What are the main environmental differences between South and Western Russia? And the type of soil?
Line 104 - We can find in the literature many papers where Bacillus velezensis is used in the biocontrol of different genera of phytopathogenic fungi in crops of high agricultural importance such as potato, mango, avocado, papaya, citrus, tomato, blueberry, blackberry, raspberry, zucchini, melon, cucumber, watermelon and others (Galindo et al., 2013; Balderas-Ruíz et al., 2020; Cui et al., 2020; and others). Therefore, your statement is not true.
Line 108 – what do you mean with much more severe climate conditions? In table 1, the temperature is presented in Celsius degree varying from 17.2 to 20.2. These are not low temperatures. And the rest of the year? How much do the climate conditions of the rest of the year influence the growing season?
Materials and methods
Line 118 - potato-glucose agar - Inform the formula or supplier.
Line 119 - For each experiment, fresh liquid cultures of the strains were prepared on the original optimized nutrient medium – How can we know what is the original optimized nutrient medium?
Line 121 – why the strains were grown in different temperatures and time of incubation?
Line 127 – What were the conditions of Western Siberia?
Line 129 - How were the cell numbers determined? The experiment was carried out by the agar blocks techniques. Provide a reference.
Line 133 – media instead of medium.
Line 144 - The average temperature of these months of 2015 was 18.6 °C, and in 2016 - 19.1 °C. Are these temperatures considered low ones?
Data presented in Table 1 can be shown here in the text. Table 1 is not necessary.
Line 159 - During the growing season – how long did it take? How many measurements?
Line 161 - At each site of the field, 10 plants were counted in four replications. Site of the field – Do you mean the plot of 30 m2? – counted for what? Four replications of what? Please, clarify.
Tables 2 and 3 – the colony diameters – show average with standard deviations.
Line 225 - the inhibitory effect was not so much – was not so evident? Please clarify.
Table 4 and the following ones – substitute Bacillus subtilis for B. velezensis
Line 231 - the disease incidence in both years of research was at least 80% on two cultivars in control (Table 4). – something is missing in this sentence.
Line 234 - and weather conditions of the year…. How did you conclude that?
Line 236 - in all cases except the strain BZR 517 on the cv. Svitanok kievskiy (P> 0.05) (Table 4). How can we see this result in Table 4? In all columns, data are statistically different.
Line 239 – these data are not shown in table 4.
Line 248 - In 2016, with less air humidity, the effect of bacilli on the disease incidence and severity was statistically less than in the previous year (Table 4). Which evidence you have to correlate humidity with the results obtained?
Line 256 - In daughter tubers of cv. Yuna, the results for both strains were similar in 2015 (Table 4). However, in 2016, a greater value was obtained for the strain BZR 517 (P <0.05) (Table 4). Are these results correct? The results seem to be similar in 2016. A greater value of what? Explain
Tables 5, 6 and 7 (titles) – potato cultivars differed by disease resistance? What do you mean? In materials and methods, you presented the cultivars as two potato cultivars of different maturity groups: early maturity cv. Yuna and mid-early cv. Svitanok kievskiy. Do they naturally differ in disease resistance? Or there is a problem in the titles?
Table 5 – standard deviations are important to be presented.
Overall, in 10 weeks after planting, the treatment with both strains for two years on two potato cultivars led to a significant increase of morphometric characteristics of plants (Table 5). - Also in Yuna with BZR 517?
Table 6 – How to explain the results observed with Yuna in 2016, mainly in the medium fraction?
Can Table 7 be incorporated in Table 6? Also add standard deviations.
Line 285 - due to the weather conditions. How this correlation was done? What happened?
Line 295 - both B. velezensis strains significantly reduced the disease severity on cv. Kirzhach (P<0.05), the difference between 336 g and 517 strains was not statistically significant (P>0.05). Please rewrite to make the sentence clear.
Line 309 - This fact may be related both to the different susceptibility of cvs. Kirzhach and Kolokolchik, and to the different weather conditions in 2015 and 2016. This statement is quite speculative.
Discussion
Line 328 - However, no such experiments have been conducted using B. velezensis. Minimize affirmative.
Line 333 - This showed the antagonistic activity of B. velezensis strains against R. solani in the region differs by environmental conditions from the location where they were isolated. Please correct the English language.
Line 340 - the dependence of the bacilli efficacy on abiotic factors? Please explain. This is in accordance with data that some Bacillus strains offer additional benefit to the plants, such as more tolerance to abiotic stress, and growth promotion. I agree with the last sentence but it has nothing to do with the previous one, unless the meaning of the phrase is not that.
Line 345 - two B. subtilis strains – B. velezensis
Line 351 - that less than usually applied by researchers in other localities. Do you mean that increasing the amount of bacteria you could have observed a better result?
Line 357 - the strain BZR 336 g contributed more significantly to this value. Which value?
Line 380 - this was confirmed by their influence on crop yields. Considering ISR is merely speculative in this case.
Line 383 - are able to realize their potential – what does this mean?
Line 410 - For the first time, we demonstrated B. velezensis strains potential for use as biological control agents on potato and on red raspberry. Although the results are important, other studies have already been performed using B. velezensis strains in the biological control of potato and red raspberry, as mentioned before.

Reviewer 3 ·

Basic reporting

This is a solid research demonstrated B. velezensis strains as biological control agents against R. solani on potato, and against D. applanata on red raspberry.

Except for the writing that need to be improved, the research was solid and very well organized and designed.

Experimental design

Line 158, Before planting, potato seed 158tubers were dipped for a few minutes in a suspension of the strains BZR 336g and BZR 517159(106 CFU mL-1). Please specify how many minutes for the coating (dipping)?

Validity of the findings

Please present Figure 1S as Figure 1 (not as supplemental figure).

Additional comments

The researches and finding are very interesting. However, the writing could be organized and presentated in a better way, for instance, line 321 to 328 could be moved to introduction section.
This just one example, the discussion part needs to be reorganized.

Ideally, please present some images showing the field crop growth and disease symptom (control/suppression).

---

## Round 0.2 · Minor Revisions

Dear Authors, Please note that Reviewer 2 still has some minor points that must be addressed before your article could be accepted for publication. I believe these are minor points that must and can be addressed rapidly. Please take care of this as soon as possible and then re-submit

Reviewer 1 ·

Basic reporting

Clear and unambiguous, professional English used throughout.

Experimental design

Research question well defined, relevant & meaningful.

Validity of the findings

Ok

Additional comments

No comment

Reviewer 2 ·

Basic reporting

In general, the authors answered accordingly to the requests made during the review. However, I have still a few comments presented here.

Experimental design

Methods are now described with more details.

Validity of the findings

The findings are valid and contribute to the research field.

Additional comments

In general, the authors answered accordingly to the requests made during the review. However, I have still a few comments:
1. I do believe that the text would benefit after being corrected by a native English specialized professional.
2. The authors mention in the abstract that “for the first time, they demonstrated B. velezensis strains potential for use as biological control agents against R. solani on potato”; in the introduction section that “no other research concerning biocontrol of R. solani in potato”; and in conclusion section “For the first time, we demonstrated B. velezensis strains potential for use as biological control agents against R. solani on potato”. All these statements have to be minimized. There are other studies already published demonstrating the effect of Bacillus velezensis against R. solani on potato. Only one example: Cui et al. (2020), Biological Control 141, 104156.
3. Table 4. Effect of B. subtilis strains on morphometric characteristics – although you stated that the title has been corrected, Bacillus subtilis is already in the title.
4. Note in Table 6 – please correct.
5. Figure 1 – title should be presented in English
6. Charon-Lamoureux V, Thérien M, Konk A, Beauregard PB. 2020. Bacillus subtilis and Bacillus velezensis population dynamic and quantification of 2 spores after… correct: quantification of spores

---

## Round 0.3 · accepted · Accept

Thank you for submitting to PeerJ.